# DAYLIGHT: ASSESSING GENERALIZATION SKILLS OF DEEP REINFORCEMENT LEARNING AGENTS

## ABSTRACT

Deep reinforcement learning algorithms have recently achieved significant success in learning high-performing policies from purely visual observations. The ability to perform end-to-end learning from raw high dimensional input alone has led to deep reinforcement learning algorithms being deployed in a variety of fields. Thus, understanding and improving the ability of deep reinforcement learning agents to generalize to unseen data distributions is of critical importance. Much recent work has focused on assessing the generalization of deep reinforcement learning agents by introducing specifically crafted adversarial perturbations to their inputs. In this paper, we propose another approach that we call *daylight*: a framework to assess the generalization skills of trained deep reinforcement learning agents. Rather than focusing on worst-case analysis of distribution shift, our approach is based on black-box perturbations that correspond to semantically meaningful changes to the environment or the agent's visual observation system ranging from brightness to compression artifacts. We demonstrate that even the smallest changes in the environment cause the performance of the agents to degrade significantly in various games from the Atari environment despite having orders of magnitude lower perceptual similarity distance compared to state-of-the-art adversarial attacks. We show that our framework captures a diverse set of bands in the Fourier spectrum, giving a better overall understanding of the agent's generalization capabilities. We believe our work can be crucial towards building resilient and generalizable deep reinforcement learning agents.

## 1 INTRODUCTION

Following the initial work of Mnih et al. (2015), the use of DNNs as function approximators in reinforcement learning has led to a dramatic increase in the capabilities of RL agents Schulman et al. (2017); Lillicrap et al. (2015). In particular, these developments allow for the direct learning of strong policies from raw, high-dimensional inputs such as visual observations. With the successes of these new methods come new challenges regarding the robustness and generalization capabilities of deep RL agents.

One line of research has focused on the high sensitivity of deep neural networks to imperceptible, adversarial perturbations to visual inputs, first in the setting of image classification Szegedy et al. (2014); Goodfellow et al. (2015) and more recently for deep reinforcement learning Huang et al. (2017); Kos & Song (2017). Since one of the main reasons for the success and popularity of deep RL is the ability to learn directly from visual observations alone, this non-robustness to small adversarial perturbations is a serious concern Chokshi (2020); Vlasic & Boudette (2016); Kunkle (2018). However, existing adversarial formulations for deep reinforcement learning require high computational effort to produce the perturbations, knowledge of the network used to train the agent, knowledge of the environment, real-time access to and manipulation of the agent's state observations.

In this paper, we propose a more realistic scenario where we do not have access to any of the above, and the adversary essentially consists of realistic changes in the natural environment or in the agent's observation system. For instance, if our deep reinforcement learning agent is operating a self-driving car one could plausibly expect changes in daylight levels, shifts in angle due to terrain, fog on the camera lens, or compression artifacts from the camera processor. We believe that our proposed framework is semantically more meaningful than arbitrary $\ell_p$-norm bounded pixel perturbations.

Prior work on image classification Dodge & Karam (2016) showed that image quality distortions can reduce the accuracy of DNN classfiers. Moreover, recent work by Ford et al. (2019) showed that while adversarial training for image classifiers reduced their vulnerability towards perturbations corresponding to high frequency in the Fourier domain, it actually made the models more vulnerable to low frequency perturbations including fog and contrast changes. Therefore, it is important to investigate model robustness and generalization throughout different bands in the frequency domain. We believe that being able to accurately assess the generalization capabilities of deep reinforcement learning agents is an initial step towards building robust and reliable agents. For these reasons, in this work we investigate the robustness of trained deep reinforcement learning agents and make the following contributions:

- We propose a realistic threat model called *daylight* and a generalization framework for deep reinforcement learning agents that aims to assess the robustness of the agents to basic environmental and observational changes.

- We run multiple experiments in the Atari environment in various games to demonstrate the degradation in performance of deep reinforcement learning agents.

- We compare our threat model with the state-of-the-art adversarial method based on $\ell_p$-norm changes, and we show that our daylight framework results in competitive, and almost always larger impact, with lower perceptual similarity distance.

- We evaluate the daylight framework in the time domain and show that several works based on the timing perspective of adversarial formulations can be revisited within our daylight framework.

- Finally, we investigate the frequency domain of our framework and state-of-the-art targeted attacks. We show that our framework captures different bands of the frequency spectrum, thus yielding a better estimate of the model robustness.

## 2 RELATED WORK

### 2.1 PRELIMINARIES

In this paper we consider Markov Decision Processes (MDPs) given by a tuple $(\mathbb{S}, \mathbb{A}, P, r, \gamma, s_0)$. The reinforcement learning agent interacts with the MDP by observing states $s \in \mathbb{S}$, and then taking actions $a \in \mathbb{A}$. The probability of transitioning to state $s'$ when the agent takes action $a$ in state $s$ is determined by the transition probability $P : \mathbb{S} \times \mathbb{A} \times \mathbb{S} \to \mathbb{R}$. The reward received by the agent when taking action $a$ in state $s$ is given by the reward function $r : \mathbb{S} \times \mathbb{A} \to \mathbb{R}$. The goal of the agent is to learn a policy $\pi_\theta : \mathbb{S} \times \mathbb{A} \to \mathbb{R}$ which takes an action $a$ in state $s$ that maximizes the cumulative discounted reward $\sum_{t=0}^{T-1} \gamma^t r(s_t, a_t)$. Here $s_0$ is the initial state of the agent, and $\gamma$ is the discount factor. For deep Q networks (DQN) the optimal policy is determined by learning the state-action value function $Q(s, a)$. For a state $s$ we use $\mathcal{F}(s)$ to denote the 2D discrete Fourier transform.

### 2.2 CRAFTING ADVERSARIAL PERTURBATIONS

Szegedy et al. (2014) proposed to minimize the distance between the original image and adversarially produced image to create adversarial perturbations. The authors used box-constrained L-BFGS to solve this optimization problem. Goodfellow et al. (2015) introduced the fast gradient method (FGM)

$$x_{\text{adv}} = x + \epsilon \cdot \frac{\nabla_x J(x, y)}{||\nabla_x J(x, y)||_p}, \tag{1}$$

for crafting adversarial examples in image classification by taking the gradient of the cost function $J(x, y)$ used to train the neural network in the direction of the input, where $x$ is the input, $y$ is the output label, and $J(x, y)$ is the cost function for image classification. Carlini & Wagner (2017) introduced targeted attacks in the image classification domain based on distance minimization between the adversarial image and the original image while targeting a particular label. In the deep reinforcement learning domain the Carlini & Wagner (2017) formulation is

$$\min_{s_{\text{adv}} \in \mathbb{S}} \quad \|s_{\text{adv}} - s\|_p$$
$$\text{subject to} \quad a^*(s) \neq a^*(s_{\text{adv}}),$$

where $s$ is the unperturbed input, $s_{\text{adv}}$ is the adversarially perturbed input, $a^*(s)$ is the action taken in the unperturbed state, and $a^*(s_{\text{adv}})$ is the action taken in the adversarial state. This formulation attempts to minimize the distance to the original state, constrained to states leading to sub-optimal actions as determined by the $Q$-network. In contrast to adversarial attacks, in our proposed threat model we will not need any information on the cost function used to train the network, $Q$-network of the trained agent or the visited states themselves.

## 2.3 ADVERSARIAL APPROACH IN DEEP REINFORCEMENT LEARNING

The first adversarial attacks on deep reinforcement learning introduced by Huang et al. (2017) and Kos & Song (2017) adapted FGSM from image classification to the deep RL setting. Subsequently, Mandlekar et al. (2017) used FGSM perturbations for adversarial training of deep RL agents. Pinto et al. (2017); Gleave et al. (2020) focused on modeling the interaction between adversary and the agent, while Lin et al. (2017); Sun et al. (2020) focused on strategically timing when (i.e. in which state) to attack an agent using perturbations computed with the Carlini & Wagner (2017) adversarial formulation.

## 2.4 PERCEPTUAL SIMILARITY

Zhang et al. (2018) found that internal activations of networks trained for high-level tasks correspond to human perceptual judgements across different network architectures Iandola et al. (2016), Krizhevsky et al. (2012), Simonyan & Zisserman (2015) without calibration. Furthermore, the authors propose a method to measure the perceptual distance between two images with the Learned Perceptual Image Patch Similarity (LPIPS) metric. We compare the distance between adversarial states $s_{\text{adv}}$ and the original states $s$ with the LPIPS metric. We refer to the LPIPS metric as $\mathcal{P}_{\text{similarity}}$ throughout the paper. $\mathcal{P}_{\text{similarity}}(s, s_{\text{adv}})$ returns the distance between $s$ and $s_{\text{adv}}$ based on network activations. Zhang et al. (2018) show that $\mathcal{P}_{\text{similarity}}$ results in a reliable approximation of human perception.

## 2.5 IMPACT

We define the normalized impact of an adversary on the agent as,

$$\mathcal{I} = \frac{\text{Score}_{\text{max}} - \text{Score}_{\text{adv}}}{\text{Score}_{\text{max}} - \text{Score}_{\text{min}}}. \tag{2}$$

$\text{Score}_{\text{max}}$ is the score at the end of the episode achieved by the agent who takes the action that maximizes its $Q(s, a)$ function in every state visited, and $\text{Score}_{\text{min}}$ is the score at the end of the episode achieved by the agent who takes the action that minimizes its $Q(s, a)$ function in every state visited. $\text{Score}_{\text{adv}}$ is the score at the end of the episode achieved by the agent who takes the action that maximizes $Q(s_{\text{adv}}, a)$ under the influence of the adversary in every state visited. We report the results in a normalized scale, because we observed the agent can collect stochastic rewards even though it chooses the action that minimizes its $Q(s, a)$ function in every state visited.

## 3 DAYLIGHT: A GENERALIZATION TESTING FRAMEWORK

In our generalization framework[1] we propose a baseline to test the robustness of trained deep reinforcement learning agents with respect to several realistic failures of the agent's observations. This is in contrast to prior work that focused on the presence of a strong adversary with prior access to training details of the agent's neural network, real time access to agent's perception system, and highly computationally demanding adversarial formulations used to compute simultaneous perturbations. In our model we consider the environment itself as an adversary and we examine several environmental

---

[1]https://daylightframework.github.io

changes such as: changes in the brightness of the environment, blurring of the observation, rotation of the observation, perspective transformation, shifting, and compression artifacts. These changes from our model can be easily linked to naturally occurring changes in the environment. For instance, for a self driving car a brightness change can be linked to the time of day, or the appearance of reflective objects or shadows. Rotation, perspective transformation, and shifting can be linked to driving on a road with varied terrain. Blurring can be linked to a rainy day, foggy weather or a fogged up camera lens utilized by the agent. In the remainder of this section we compare the impact values and perceptual similarities of the daylight framework with the state-of-the-art targeted adversarial attack proposed by Carlini & Wagner (2017).

## 3.1 BRIGHTNESS AND CONTRAST

The first component of our framework focusing on testing the trained agents in different brightness and contrast levels using a linear brightness and contrast transformation,

$$s_{\text{adv}}(i, j) = s(i, j) \cdot \alpha + \beta, \tag{3}$$

where $s(i, j)$ is the $ij^{\text{th}}$ pixel of state $s$, and $\alpha$ and $\beta$ are the linear brightness parameters. In Table 1 we show the impacts and perceptual similarity distances with corresponding $\alpha, \beta$ values. In all of the games except BankHeist brightness and contrast change results in higher impact. The perceptual similarity distance of brightness and contrast is lower in every game. In Figure 3 we show the corresponding states $s$ and $s_{\text{adv}}$ for all of the *daylight* framework.

Table 1: Impacts of Carlini & Wagner (2017) and brightness & contrast (B&C) with corresponding perceptual similarity $\mathcal{P}_{\text{similarity}}$, and the $[\alpha, \beta]$.

| Games | C&W Impact | B&C Impact | C&W $\mathcal{P}_{\text{similarity}}$ | B&C $\mathcal{P}_{\text{similarity}}$ | $[\alpha, \beta]$ |
|---|---|---|---|---|---|
| BankHeist | 0.986±0.009 | 0.971± 0.030 | 0.171 | 0.127 | [1.2,40] |
| JamesBond | 0.516±0.231 | 0.896±0.047 | 0.326 | 0.015 | [0.9,20] |
| Pong | 0.995±0.014 | 0.996±0.009 | 0.608 | 0.245 | [1.7,40] |
| RiverRaid | 0.9498±0.030 | 0.973±0.016 | 0.255 | 0.168 | [2.4,-275] |
| TimePilot | 0.774±0.159 | 0.906±0.239 | 0.279 | 0.070 | [2.4,-260] |

## 3.2 BLURRING

The second component in our *daylight* framework is blurring. *Median bluring* is a nonlinear noise removal technique that replaces the original pixel value with the median pixel value of its neighbouring pixels. A kernel size $k$ means that the median is computed over a $k \times k$ neighborhood of the original pixel. In Table 2 we show the impact values and perceptual similarity distance with corresponding kernel size. Only in BankHeist do we observe that the impacts and perceptual similarity distance are comparable. For the rest of the games impact is higher and perceptual similarity distance is lower for blurring.

Table 2: Impacts of Carlini & Wagner (2017) and blurring with corresponding perceptual similarity $\mathcal{P}_{\text{similarity}}$, and kernel size.

| Games | C&W Impact | Blurring Impact | C&W $\mathcal{P}_{\text{similarity}}$ | Blurring $\mathcal{P}_{\text{similarity}}$ | Kernel Size |
|---|---|---|---|---|---|
| BankHeist | 0.986±0.009 | 0.983±0.009 | 0.171 | 0.168 | 5 |
| JamesBond | 0.516±0.231 | 0.634±0.200 | 0.326 | 0.048 | 3 |
| Pong | 0.995±0.014 | 1.0±0.000 | 0.245 | 0.098 | 3 |
| RiverRaid | 0.9498±0.030 | 0.968±0.015 | 0.255 | 0.134 | 5 |
| TimePilot | 0.774±0.159 | 0.805±0.150 | 0.279 | 0.124 | 5 |

## 3.3 ROTATION

The next component in our *daylight* framework is rotation. In Table 3 we show impact values and perceptual similarity distance with corresponding rotation angle. In all of the games except Pong rotation results in higher impact and orders of magnitude lower perceptual similarity distance. In Pong the impact is comparable and the perceptual similarity distance is lower by a factor of 2.

Table 3: Impacts of Carlini & Wagner (2017) and rotation with corresponding perceptual similarity $\mathcal{P}_{\text{similarity}}$, and the rotation angle (RD denotes rotation degree).

| Games | C&W Impact | Rotation Impact | C&W $\mathcal{P}_{\text{similarity}}$ | Rotation $\mathcal{P}_{\text{similarity}}$ | RD |
|---|---|---|---|---|---|
| BankHeist | 0.986±0.009 | 1.0±0.004 | 0.171 | 0.064 | 1.4 |
| JamesBond | 0.516±0.231 | 0.725±0.189 | 0.326 | 0.029 | 1.6 |
| Pong | 0.995±0.014 | 0.99±0.015 | 0.245 | 0.112 | 3 |
| RiverRaid | 0.9498±0.030 | 0.965±0.042 | 0.255 | 0.059 | 1.8 |
| TimePilot | 0.774±0.159 | 0.910±0.158 | 0.279 | 0.068 | 5 |

## 3.4 SHIFTING

The next component in our *daylight* framework is shifting. Shifting an image moves the elements of the image matrix along any dimension by any number of elements. In this subsection we will shift the inputs in the $x$ or $y$ direction with as few pixels shifted as possible. We use $[t_i, t_j]$ to denote the distance shifted, where $t_i$ is in the direction of $x$ and $t_j$ is in the direction of $y$. In Table 4 we show the impact values and perceptual similarity distances for both Carlini & Wagner (2017) and shifting. For all of the games shifting yields higher impact and lower perceptual similarity distance.

Table 4: Impacts of Carlini & Wagner (2017) and shifting with corresponding perceptual similarity $\mathcal{P}_{\text{similarity}}$, and the shifting $[t_i, t_j]$.

| Games | C&W Impact | Shifting Impact | C&W $\mathcal{P}_{\text{similarity}}$ | Shifting $\mathcal{P}_{\text{similarity}}$ | $[t_i, t_j]$ |
|---|---|---|---|---|---|
| BankHeist | 0.986±0.009 | 0.989±0.005 | 0.171 | 0.060 | [1,1] |
| JamesBond | 0.516±0.231 | 0.989±0.140 | 0.326 | 0.048 | [0,1] |
| Pong | 0.995±0.014 | 1.0±0.00 | 0.608 | 0.217 | [2,1] |
| RiverRaid | 0.9498±0.030 | 0.9568±0.023 | 0.255 | 0.095 | [1,2] |
| TimePilot | 0.774±0.159 | 0.851±0.199 | 0.279 | 0.120 | [2,2] |

## 3.5 COMPRESSION ARTIFACTS

In this section we look at jpeg compression artifacts caused by the discrete cosine transform (DCT) resulting in the loss of high frequency components (ringing and blocking). In Table 5 we show the impact values and perceptual similarities of Carlini & Wagner (2017) and compression artifacts (CA).

Table 5: Impacts of Carlini & Wagner (2017) and compression artifacts (CA) with corresponding perceptual similarity $\mathcal{P}_{\text{similarity}}$.

| Games | C&W Impact | CA Impact | C&W $\mathcal{P}_{\text{similarity}}$ | CA $\mathcal{P}_{\text{similarity}}$ |
|---|---|---|---|---|
| BankHeist | 0.986±0.009 | 0.984±0.013 | 0.171 | 0.067 |
| JamesBond | 0.516±0.231 | 1.0±0.128 | 0.326 | 0.035 |
| Pong | 0.995±0.014 | 0.962±0.032 | 0.608 | 0.029 |
| RiverRaid | 0.9498±0.030 | 0.8218±0.051 | 0.255 | 0.057 |
| TimePilot | 0.774±0.159 | 0.790±0.271 | 0.279 | 0.067 |

Only in Pong and Riverraid do we observe a lower impact than Carlini & Wagner (2017) while the perceptual similarity distance was orders of magnitude smaller for compression artifacts. In BankHeist the impact is comparable, and in the rest of the games compression artifacts result in higher impact and lower perceptual similarity distance compared to Carlini & Wagner (2017).

### 3.6 PERSPECTIVE TRANSFORMATION

The final component of our *daylight* framework is perspective transformation. Given four points in the plane defining a convex quadrangle, there is a unique perspective transformation mapping the corners of the square to these four points (see Equation 5 and Equation 6). We define the norm of a perspective transformation as the maximum distance that one of the corners of the square moves under this mapping. In Table 6 we show impact values and perceptual similarity distance with respect to perspective norms. For all the games we observe perspective transformation yields higher impact and lower perceptual similarity distance.

Table 6: Impacts of Carlini & Wagner (2017) and perspective transformation (PT) with corresponding perceptual similarity $\mathcal{P}_{\text{similarity}}$, and the perspective norm.

| Games | C&W Impact | PT Impact | C&W $\mathcal{P}_{\text{similarity}}$ | PT $\mathcal{P}_{\text{similarity}}$ | Perspective Norm |
|---|---|---|---|---|---|
| BankHeist | 0.986±0.009 | 1.0±0.003 | 0.171 | 0.022 | 1 |
| JamesBond | 0.516±0.231 | 0.978±0.087 | 0.326 | 0.007 | 1 |
| Pong | 0.995±0.014 | 0.996±0.009 | 0.608 | 0.029 | 3 |
| RiverRaid | 0.9498±0.030 | 0.99±0.006 | 0.255 | 0.046 | 2 |
| TimePilot | 0.774±0.159 | 0.852±0.198 | 0.279 | 0.029 | 3 |

## 4 FOURIER DOMAIN

Ford et al. (2019) showed DNN models are robust to certain bands of perturbations in the frequency domain. Furthermore, they showed that adversarial training shifts the vulnerability from high frequency noise towards low frequency noise. Moreover, Yin et al. (2019) claim that a framework that aims to measure robustness and generalization needs to firmly encapsulate different directions of the spectrum in the frequency domain. In this section we show that the *daylight* framework indeed captures a broader set of directions in the frequency domain.

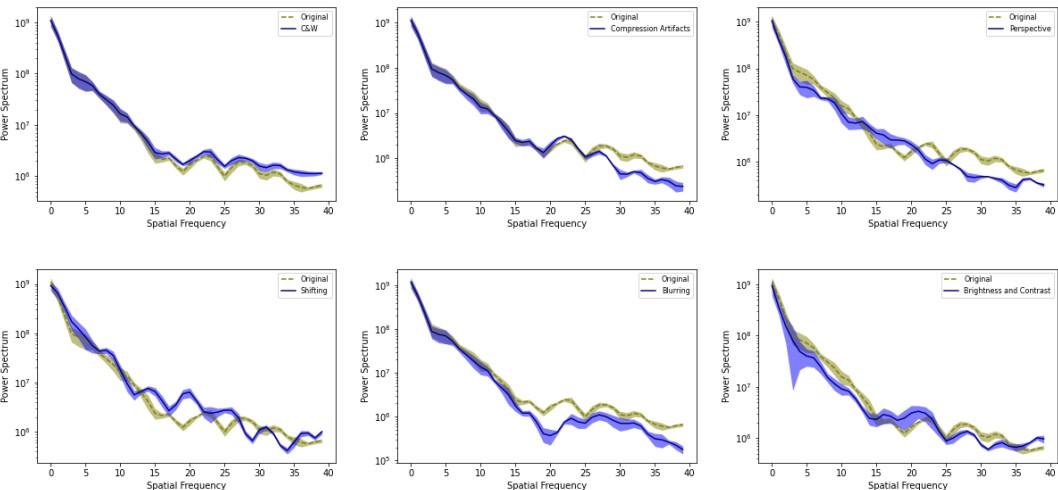

Figure 1: Riverraid power spectrum change with various perturbations: Carlini & Wagner, compression artifacts, brightness and contrast, perspective transformation, shifting, rotation.

In Figure 2 we show the Fourier spectrum of the original state $s$ and the perturbed states $s_{\text{adv}}$ from the *daylight* framework and Carlini & Wagner (2017). In Figure 1 we show the power spectral density of the original state compared to several perturbations from our proposed daylight framework and Carlini & Wagner (2017). In Figure 1 we observe that while Carlini & Wagner (2017) increases the higher frequencies, compression artifacts decrease the magnitude of the higher frequency band.

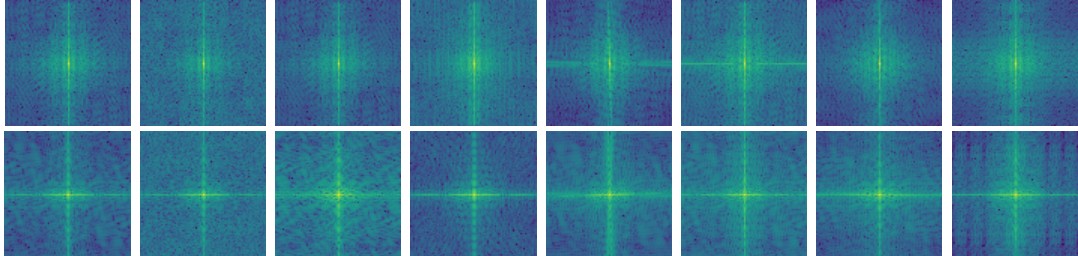

Figure 2: Rows: $\mathcal{F}(s)$ for BankHeist, $\mathcal{F}(s)$ Riverraid. Columns: original state, Carlini & Wagner, brightness and contrast, blurring, rotation, shifting, perspective tansformation, compression artifacts.

On the other hand, brightness and contrast decreases the magnitude of the low frequency band, and shifting increases the midband. Blurring decreases the midband and high frequencies together, and perspective transformation decreases the low frequecies and high frequencies while increasing the midband. We believe capturing the susceptibilities towards perturbations in different bands of the frequency domain is a key step towards building robust agents.

## 5    TIMING PERSPECTIVE

In the previous sections we tested our agents generalization capabilities in modified environments with our initial thread model. This modification to the environment was applied to every state that the agent visited for both Carlini & Wagner (2017) and our *daylight* framework. In this section we will examine the effect of both adversarial models when the perturbations are applied to only a small fraction of states. For this purpose we introduce the adversarial states $s_{\mathrm{adv}}$ in randomly sampled states where the observation $s_{\mathrm{adv}}$ is observed by the agent with probability $p$, and the original states $s$ is observed observed by the agent with probability $1 - p$. We use $n_{s_{\mathrm{adv}}}$ to denote the number of states where the agent observed $s_{\mathrm{adv}}$ instead of the original state $s$, and we use $n_s$ to denote the total number of states visited by the agent in the given episode. We use $e_{\mathrm{adv}}$ to denote the fraction $n_{s_{\mathrm{adv}}}/n_s$ of adversarial perturbations per episode.

Table 7: Impact comparison with the fraction of adversarial observation probability $p$.

|  | RiverRaid | TimePilot | BankHeist | Pong | JamesBond |
|---|---|---|---|---|---|
| C&W Impact | 0.359 | 0.148 | 0.249 | 0.077 | 0.021 |
| Shifting Impact | 0.513 | 0.374 | 0.326 | 0.114 | 0.165 |
| Perspective Transformation Impact | 0.391 | 0.315 | 0.338 | 0.108 | 0.121 |
| Blurring Impact | 0.501 | 0.155 | 0.304 | 0.12 | 0.319 |
| Brightness & Contrast Impact | 0.517 | 0.188 | 0.313 | 0.098 | 0.154 |
| Rotation Impact | 0.417 | 0.192 | 0.260 | 0.079 | 0.044 |
| Compression Artifacts Impact | 0.184 | 0.262 | 0.267 | 0.017 | 0.198 |
| $s_{\mathrm{adv}}$ observation probability $p$ | 0.1 | 0.02 | 0.02 | 0.06 | 0.08 |

In Table 7 we show the attack impacts of Carlini & Wagner (2017) and daylight framework with corresponding adversarial observation probability $p$ averaged over 10 random episodes. See Appendix A.4 for more details. Even for low $p$ values our proposed daylight framework obtains higher impact. Thus, to capture a broader view on the robustness of the agent, the prior work on the timing perspective by Sun et al. (2020); Lin et al. (2017) based on worst-case distributional shift, can be revisited with our *daylight* framework.

## 6    EXPERIMENTS

In our experiments we trained our agents with DDQN Wang et al. (2016) in the OpenAI Gym Brockman et al. (2016) Atari environment Bellemare et al. (2013). We test trained agents from several Atari environments averaged over 10 episodes. In Figure 3 we show the original states

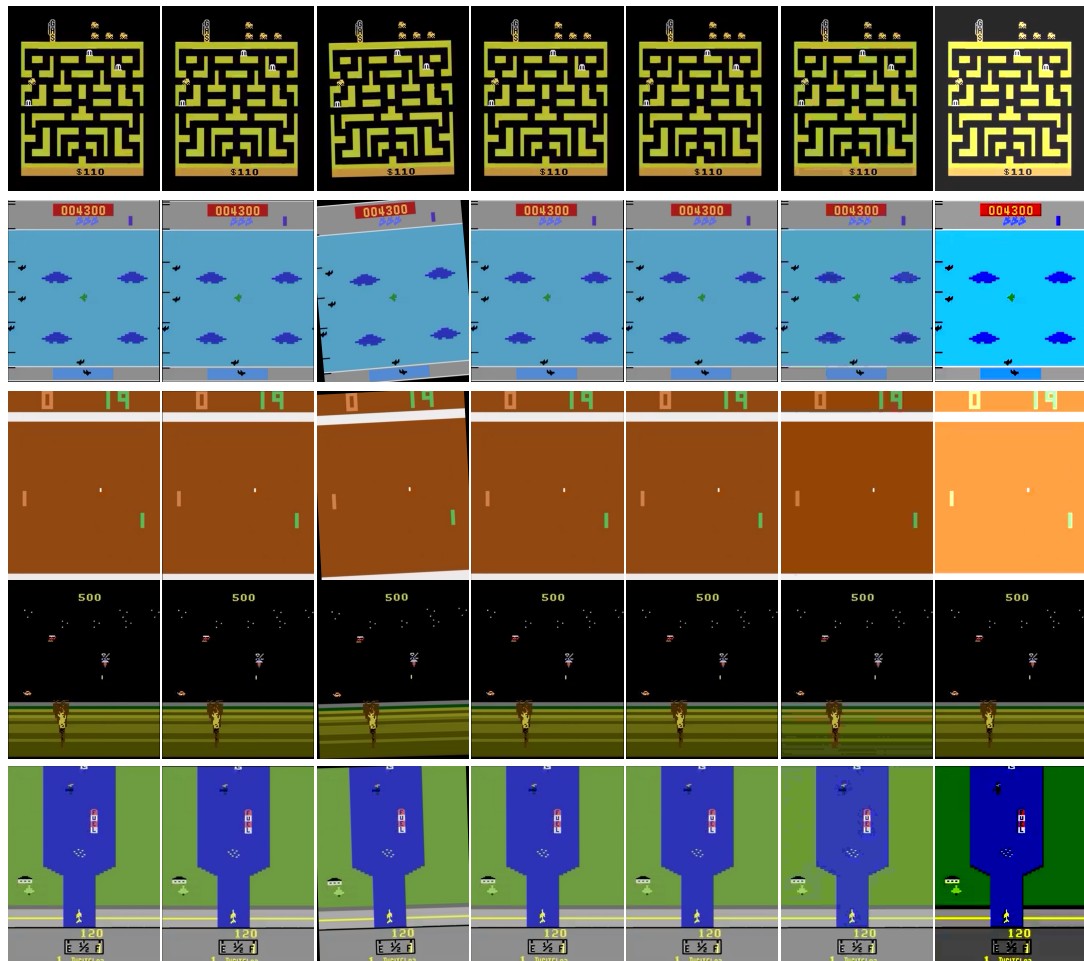

Figure 3: Original frame and environmental modifications. Columns: original frame, shifting, rotation, perspective transformation, blurring, compression artifacts. brightness and contrast. Rows: Bankheist, Timepilot, Pong, JamesBond, Riverraid.

and the environmental modifications. Interestingly, we found that a majority of the games have high robustness against rotation. On the other hand, shifting and perspective transformation can reach a higher impact level than the state-of-the-art targeted attack while not being recognizable by human perception. We observed that in some games, such as Pong and Riverraid, brightness and contrast requires radical changes to cause the agent to fail, while for others the change required is imperceptible. Another thing we observed is that for games like Pong, which is relatively trivial compared to other games in the Atari environment, the threshold values for the environmental modification are higher. When the complexity in the game increases the environment modification thresholds decrease drastically. We think that this issue could become more important as deep reinforcement learning agents are deployed in more complex and realistic scenarios.

## 7 CONCLUSION

In this paper we studied a realistic threat model based on basic environmental changes and proposed a framework called *daylight* to asses the generalization capabilities of deep reinforcement learning agents. We compared our daylight framework with the state-of-the-art adversarial attacks in the Atari environment. We demonstrated that our framework achieves higher impact on agent performance with lower perceptual similarity distance without having access to agents training details, real time access

to agents memory and perception system, and computationally demanding adversarial formulations to compute simultaneous perturbations. We investigated perturbations in the time domain and showed that the studies based on imperceptible perturbations can be revisited within the daylight framework. Finally, we show that each component of our framework contains distinct bands in the frequency domain, resulting in a better estimate of the generalization capabilities of trained agents. We believe our framework can be instrumental towards generalization and robustification of deep reinforcement learning algorithms.

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

# A APPENDIX

## A.1 RAW SCORE RESULTS

In this section we provide the raw scores of the agents under the observation modifications from both the state-of-the-art adversarial attack and the Daylight framework with components: Brightness & Contrast, Blurring, Rotation, Shifting, Compression Artifacts, and Perspective Transform .

Table 8: Raw Scores of Carlini & Wagner (2017) (C&W) and brightness & contrast (B&C) with corresponding perceptual similarity $\mathcal{P}_{\text{similarity}}$, and the $[\alpha, \beta]$.

| Games | C&W Raw Scores | B&C Raw Scores | C&W $\mathcal{P}_{\text{similarity}}$ | B&C $\mathcal{P}_{\text{similarity}}$ | $[\alpha, \beta]$ |
|---|---|---|---|---|---|
| BankHeist | 15.0 | 17.0 | 0.171 | 0.127 | [1.2,40] |
| JamesBond | 285.0 | 45.0 | 0.326 | 0.015 | [0.9,20] |
| Pong | -20.8 | -21.0 | 0.608 | 0.245 | [1.7,40] |
| RiverRaid | 1168.0 | 744.0 | 0.255 | 0.168 | [2.4,-275] |
| TimePilot | 4090.0 | 3180.0 | 0.279 | 0.070 | [2.4,-260] |

Table 9: Raw Scores of Carlini & Wagner (2017) (C&W) and blurring with corresponding perceptual similarity $\mathcal{P}_{\text{similarity}}$, and kernel size.

| Games | C&W Raw Scores | Blurring Raw Scores | C&W $\mathcal{P}_{\text{similarity}}$ | Blurring $\mathcal{P}_{\text{similarity}}$ | Kernel Size |
|---|---|---|---|---|---|
| BankHeist | 15.0 | 18.0 | 0.171 | 0.168 | 5 |
| JamesBond | 285.0 | 190.0 | 0.326 | 0.048 | 3 |
| Pong | -20.8 | -21.0 | 0.245 | 0.098 | 3 |
| RiverRaid | 1168.0 | 820.0 | 0.255 | 0.134 | 5 |
| TimePilot | 4090.0 | 3880.0 | 0.279 | 0.124 | 5 |

Table 10: Raw Scores of Carlini & Wagner (2017) (C&W) and rotation with corresponding perceptual similarity $\mathcal{P}_{\text{similarity}}$, and the rotation angle (RD denotes rotation degree).

| Games | C&W Raw Scores | Rotation Raw Scores | C&W $\mathcal{P}_{\text{similarity}}$ | Rotation $\mathcal{P}_{\text{similarity}}$ | RD |
|---|---|---|---|---|---|
| BankHeist | 15.0 | 2.0 | 0.171 | 0.064 | 1.4 |
| JamesBond | 285.0 | 190.0 | 0.326 | 0.029 | 1.6 |
| Pong | -20.8 | -20.6 | 0.245 | 0.112 | 3 |
| RiverRaid | 1168.0 | 873.0 | 0.255 | 0.059 | 1.8 |
| TimePilot | 4000.0 | 3150.0 | 0.279 | 0.068 | 5 |

Table 11: Raw Scores of Carlini & Wagner (2017) (C&W) and shifting with corresponding perceptual similarity $\mathcal{P}_{\text{similarity}}$, and the shifting $[t_i, t_j]$.

| Games | C&W Raw Scores | Shifting Raw Scores | C&W $\mathcal{P}_{\text{similarity}}$ | Shifting $\mathcal{P}_{\text{similarity}}$ | $[t_i, t_j]$ |
|---|---|---|---|---|---|
| BankHeist | 15.0 | 13.0 | 0.171 | 0.060 | [1,1] |
| JamesBond | 285.0 | 70.0 | 0.326 | 0.048 | [0,1] |
| Pong | -20.8 | -21.0 | 0.608 | 0.217 | [2,1] |
| RiverRaid | 1168.0 | 988.0 | 0.255 | 0.095 | [1,2] |
| TimePilot | 4000.0 | 3560.0 | 0.279 | 0.120 | [2,2] |

Table 12: Raw Scores of Carlini & Wagner (2017) (C&W) and compression artifacts (CA) with corresponding perceptual similarity $\mathcal{P}_{\text{similarity}}$.

| Games | C&W Raw Scores | CA Raw Scores | C&W $\mathcal{P}_{\text{similarity}}$ | CA $\mathcal{P}_{\text{similarity}}$ |
|---|---|---|---|---|
| BankHeist | 15.0 | 17.0 | 0.171 | 0.067 |
| JamesBond | 250.0 | 60.0 | 0.326 | 0.035 |
| Pong | -20.8 | -19.4 | 0.608 | 0.029 |
| RiverRaid | 1168.0 | 2589.0 | 0.255 | 0.057 |
| TimePilot | 4090.0 | 3980.0 | 0.279 | 0.067 |

Table 13: Raw Scores of Carlini & Wagner (2017) (C&W) and perspective transformation (PT) with corresponding perceptual similarity $\mathcal{P}_{\text{similarity}}$, and the perspective norm.

| Games | C&W Raw Scores | PT Raw Scores | C&W $\mathcal{P}_{\text{similarity}}$ | PT $\mathcal{P}_{\text{similarity}}$ | Perspective Norm |
|---|---|---|---|---|---|
| BankHeist | 15.0 | 1.0 | 0.171 | 0.022 | 1 |
| JamesBond | 285.0 | 75.0 | 0.326 | 0.007 | 1 |
| Pong | -20.8 | -20.9 | 0.608 | 0.029 | 3 |
| RiverRaid | 1168.0 | 486.0 | 0.255 | 0.046 | 2 |
| TimePilot | 4000.0 | 3550.0 | 0.279 | 0.029 | 3 |

## A.2 IMPACTS WITH FIXED SCORES

For the scope of the paper we used the Impact definition in Equation 2 when we compare our proposed Daylight framework to the state-of-the-art targeted attacks. For more generalized comparison between different algorithms and different games one can use $\text{Score}_{\text{clean}}$ as the score of the agent without any modification to agent's observations system at the end of the episode and $\text{Score}_{\text{min}}^{\text{fixed}}$ as the fixed minimum score for a given game. Thus, we define the generalized impact as,

$$\mathcal{I}_{\text{general}} = \frac{\text{Score}_{\text{clean}} - \text{Score}_{\text{adv}}}{\text{Score}_{\text{clean}} - \text{Score}_{\text{min}}^{\text{fixed}}}. \tag{4}$$

From Table 14 through Table 19 we set $\text{Score}_{\text{min}}^{\text{fixed}}$ for Bankheist 0, JamesBond 0, Pong -21, Riverraid 0, and TimePilot 0.

Table 14: Generalized Impacts of Carlini & Wagner (2017) (C&W) and brightness & contrast (B&C) with corresponding perceptual similarity $\mathcal{P}_{\text{similarity}}$, and the $[\alpha, \beta]$.

| Games | C&W $\mathcal{I}_{\text{general}}$ | B&C $\mathcal{I}_{\text{general}}$ | C&W $\mathcal{P}_{\text{similarity}}$ | B&C $\mathcal{P}_{\text{similarity}}$ | $[\alpha, \beta]$ |
|---|---|---|---|---|---|
| BankHeist | 0.982 | 0.966 | 0.171 | 0.127 | [1.2,40] |
| JamesBond | 0.451 | 0.913 | 0.326 | 0.015 | [0.9,20] |
| Pong | 0.995 | 1.0 | 0.608 | 0.245 | [1.7,40] |
| RiverRaid | 0.928 | 0.951 | 0.255 | 0.168 | [2.4,-275] |
| TimePilot | 0.567 | 0.663 | 0.279 | 0.070 | [2.4,-260] |

Table 15: Generalized Impacts of Carlini & Wagner (2017) (C&W) and blurring with corresponding perceptual similarity $\mathcal{P}_{\text{similarity}}$, and kernel size.

| Games | C&W $\mathcal{I}_{\text{general}}$ | Blurring $\mathcal{I}_{\text{general}}$ | C&W $\mathcal{P}_{\text{similarity}}$ | Blurring $\mathcal{P}_{\text{similarity}}$ | Kernel Size |
|---|---|---|---|---|---|
| BankHeist | 0.982 | 0.979 | 0.171 | 0.168 | 5 |
| JamesBond | 0.451 | 0.635 | 0.326 | 0.048 | 3 |
| Pong | 0.995 | 1.0 | 0.245 | 0.098 | 3 |
| RiverRaid | 0.928 | 0.946 | 0.255 | 0.134 | 5 |
| TimePilot | 0.567 | 0.589 | 0.279 | 0.124 | 5 |

Table 16: Generalized Impacts of Carlini & Wagner (2017) (C&W) and rotation with corresponding perceptual similarity $\mathcal{P}_{\text{similarity}}$, and the rotation angle (RD denotes rotation degree).

| Games | C&W $\mathcal{I}_{\text{general}}$ | Rotation $\mathcal{I}_{\text{general}}$ | C&W $\mathcal{P}_{\text{similarity}}$ | Rotation $\mathcal{P}_{\text{similarity}}$ | RD |
|---|---|---|---|---|---|
| BankHeist | 0.982 | 0.997 | 0.171 | 0.064 | 1.4 |
| JamesBond | 0.451 | 0.635 | 0.326 | 0.029 | 1.6 |
| Pong | 0.995 | 0.99 | 0.245 | 0.112 | 3 |
| RiverRaid | 0.928 | 0.942 | 0.255 | 0.059 | 1.8 |
| TimePilot | 0.567 | 0.581 | 0.279 | 0.068 | 5 |

Table 17: Generalized Impacts of Carlini & Wagner (2017) (C&W) and shifting with corresponding perceptual similarity $\mathcal{P}_{\text{similarity}}$, and the shifting $[t_i, t_j]$.

| Games | C&W $\mathcal{I}_{\text{general}}$ | Shifting $\mathcal{I}_{\text{general}}$ | C&W $\mathcal{P}_{\text{similarity}}$ | Shifting $\mathcal{P}_{\text{similarity}}$ | $[t_i, t_j]$ |
|---|---|---|---|---|---|
| BankHeist | 0.982 | 0.985 | 0.171 | 0.060 | [1,1] |
| JamesBond | 0.451 | 0.865 | 0.326 | 0.048 | [0,1] |
| Pong | 0.995 | 1.0 | 0.608 | 0.217 | [2,1] |
| RiverRaid | 0.928 | 0.935 | 0.255 | 0.095 | [1,2] |
| TimePilot | 0.567 | 0.623 | 0.279 | 0.120 | [2,2] |

Table 18: Generalized Impacts of Carlini & Wagner (2017) (C&W) and compression artifacts (CA) with corresponding perceptual similarity $\mathcal{P}_{\text{similarity}}$.

| Games | C&W $\mathcal{I}_{\text{general}}$ | CA $\mathcal{I}_{\text{general}}$ | C&W $\mathcal{P}_{\text{similarity}}$ | CA $\mathcal{P}_{\text{similarity}}$ |
|---|---|---|---|---|
| BankHeist | 0.982 | 0.980 | 0.171 | 0.067 |
| JamesBond | 0.451 | 0.884 | 0.326 | 0.035 |
| Pong | 0.995 | 0.962 | 0.608 | 0.029 |
| RiverRaid | 0.923 | 0.803 | 0.255 | 0.057 |
| TimePilot | 0.567 | 0.578 | 0.279 | 0.067 |

Table 19: Generalized Impacts of Carlini & Wagner (2017) (C&W) and perspective transformation (PT) with corresponding perceptual similarity $\mathcal{P}_{\text{similarity}}$, and the perspective norm.

| Games | C&W $\mathcal{I}_{\text{general}}$ | PT $\mathcal{I}_{\text{general}}$ | C&W $\mathcal{P}_{\text{similarity}}$ | PT $\mathcal{P}_{\text{similarity}}$ | Perspective Norm |
|---|---|---|---|---|---|
| BankHeist | 0.982 | 0.998 | 0.171 | 0.022 | 1 |
| JamesBond | 0.451 | 0.865 | 0.326 | 0.007 | 1 |
| Pong | 0.995 | 0.996 | 0.608 | 0.029 | 3 |
| RiverRaid | 0.928 | 0.968 | 0.255 | 0.046 | 2 |
| TimePilot | 0.567 | 0.624 | 0.279 | 0.029 | 3 |

### A.3 POLICY GRADIENT METHODS UNDER DAYLIGHT FRAMEWORK

In this section we investigate policy gradient methods. In particular, Table 20 shows the raw scores, and generalized impacts $\mathcal{I}_{\text{general}}$ of the agent trained with A3C under the Daylight framework with following observation modifications: brightness & constrast, blurring, rotation, shifting, compression artifacts and perspective transform. In Table 20 the exact same hyperparameters have been used as stated in Table 1 through Table 6 for the Daylight framework. Note that Daylight hyperparameters refers for brightnes and contrast to $[\alpha, \beta]$, for blurring to the kernel size, for rotation to rotation degree, for shifting to $[t_i, t_j]$, and for perspective transformation to perspective norm. Shifting and compression artifacts have nearly maximal impact on the performance of the agent trained with A3C, while the other perturbations all have impact at least 0.9. Note that we did not change the Daylight hyperparameters for a direct comparison between the A3C agent and the DDQN agent. Therefore, although impact is slightly lower for brightness & contrast for A3C than for DDQN, it is possible

that choosing different values of $\alpha$ and $\beta$ while minimizing the perceptual similarity $\mathcal{P}_{\text{similarity}}$ can still result in a higher impact for an agent trained with A3C.

Table 20: Raw Scores and generalized impacts of the agent trained with A3C algorithm in Pong environment and evaluated with Daylight frame work: brightness &contrast, blurring, rotation, shifting, compression artifacts (CA) and perspective transform (PT).

| Pong | Bright&Contrast | Blurring | Rotation | Shifting | CA | PT |
|---|---|---|---|---|---|---|
| Raw Scores | -17 | -20.35 | -19.96 | -20.71 | -20.89 | -19.11 |
| Generalized Impacts $\mathcal{I}_{\text{general}}$ | 0.904 | 0.984 | 0.974 | 0.993 | 0.997 | 0.954 |
| Daylight hyperparameters | [1.7,40] | 3 | 3 | [2,1] | - | 3 |

## A.4 TIMING PERSPECTIVE

Note that the fraction $e_{\text{adv}}$ can differ slightly from $p$ due to random fluctuations, therefore we also report these values in Table 7. Note that $e_{\text{adv}}$ varies between games. This was done because each game has a different minimum threshold for $e_{\text{adv}}$ to achieve stable impact across episodes.

Table 21: Impact comparison with the fraction of adversarial observations per episode $e_{\text{adv}}$.

| | RiverRaid | TimePilot | BankHeist | Pong | JamesBond |
|---|---|---|---|---|---|
| C&W Impact | 0.359 | 0.148 | 0.249 | 0.077 | 0.021 |
| Shifting Impact | 0.513 | 0.374 | 0.326 | 0.114 | 0.165 |
| Perspective Transformation Impact | 0.391 | 0.315 | 0.338 | 0.108 | 0.121 |
| Blurring Impact | 0.501 | 0.155 | 0.304 | 0.12 | 0.319 |
| Brightness & Contrast Impact | 0.517 | 0.188 | 0.313 | 0.098 | 0.154 |
| Rotation Impact | 0.417 | 0.192 | 0.260 | 0.079 | 0.044 |
| Compression Artifacts Impact | 0.184 | 0.262 | 0.267 | 0.017 | 0.198 |
| C&W $e_{\text{adv}}$ | 0.096 | 0.020 | 0.021 | 0.062 | 0.081 |
| Shifting $e_{\text{adv}}$ | 0.100 | 0.019 | 0.020 | 0.062 | 0.084 |
| Perspective Transform $e_{\text{adv}}$ | 0.098 | 0.020 | 0.020 | 0.060 | 0.082 |
| Blurring $e_{\text{adv}}$ | 0.099 | 0.019 | 0.020 | 0.061 | 0.083 |
| Brightness $e_{\text{adv}}$ | 0.101 | 0.020 | 0.018 | 0.062 | 0.082 |
| Rotation $e_{\text{adv}}$ | 0.099 | 0.021 | 0.020 | 0.061 | 0.083 |
| Compression Artifacts $e_{\text{adv}}$ | 0.097 | 0.020 | 0.020 | 0.056 | 0.080 |
| $s_{\text{adv}}$ observation probability $p$ | 0.1 | 0.02 | 0.02 | 0.06 | 0.08 |

## A.5 FORMULAS FOR PERSPECTIVE TRANSFORMATION

$$t_k \begin{bmatrix} s_i^{\text{dst}_k} \\ s_j^{\text{dst}_k} \\ 1 \end{bmatrix} = M \cdot \begin{bmatrix} s_i^{\text{src}_k} \\ s_j^{\text{src}_k} \\ 1 \end{bmatrix} \tag{5}$$

$$s_{\text{adv}}(i,j) = s \left( \frac{M_{11}s_i + M_{12}s_j + M_{13}}{M_{31}s_i + M_{32}s_j + M_{33}}, \frac{M_{21}s_i + M_{22}s_j + M_{23}}{M_{31}s_i + M_{32}s_j + M_{33}} \right) \tag{6}$$

