# OpenReview forum: "Daylight: Assessing Generalization Skills of Deep Reinforcement Learning Agents"
_ICLR.cc/2021/Conference — Reject_

### Official Review · AnonReviewer1 · 2020-10-24
**Simple attacks for reinforcement learning with pixels based on image distortions**

**Rating:** 6
**Confidence:** 4

**Review:**

################################################

Summary:
Previous work on crafting attacks for deep reinforcement learning has relied on computing adversarial examples using knowledge of the environment, policy, and optimizer. Using Atari games and DDQN, this paper shows that simple image distortions, such as brightness changes, blurring, and rotations, often has greater impact on the agent's performance and is perceptually more similar to the original images.

################################################

Pros:
1. The proposed attacks are simple, intuitively meaningful, and computationally cheap. They are black box, not requiring information about the environment and policy, which is a more realistic setting.
2. The proposed attacks seem to be better than Carlini & Wagner for most games. I was a little surprised how much, especially for JamesBond.

Cons:
1. The results are shown for only one algorithm, DDQN, only on Atari. It would be good to provide results for newer algorithms like Ape-X or other environments like DMControl.
2. Deep RL algorithms usually have large variance. It would be good to provide standard errors of the results, to accurately compare Daylight to Carlini & Wagner.
3. The writing has some weaknesses, please see below for details.

################################################

Overall: I would lean toward accepting this paper. Technically it is not very sophisticated and the proposed attacks have been considered for image classification [1], but I believe that the results have strong practical implications. That is, simple image distortions are surprisingly impactful for attacking deep RL agents.

[1] Samuel Dodge and Lina Karam. "Understanding How Image Quality Affects Deep Neural Networks". ArXiv 1604.04004.

################################################

Suggestions for writing:
1. I think it would be better if the experimental setup preceded the results in Section 3. In addition, more details should be given, such as: How were the specific games and algorithm chosen?
2. I think the paper [1] is relevant for related work, as it shows similar distortions are good attacks for image classification.
3. The notation D(s) at the bottom of page 2 is undefined.
4. The differences in Figure 2 are not very apparent. It may be helpful to add bounding boxes.

Further comments and questions:
1. What happens if we combine several of these attacks? Would that lead to even greater impact?
2. Is there any intuition for why certain games are more robust to certain image distortions than others? For example, why are the results for compression artifacts more mixed?

################################################

Update after reading other reviews and author responses:

I am happy to keep my score and remain positive about this paper; the authors have answered my questions and partially addressed my main concerns in the revised paper. Like Reviewer 3, I would hope to see complete results for A3C in the final paper.

---

> ### Author Response · Authors · 2020-11-16
> **Author Response**
>
> Thank you for your comments. We have tried to address them below. Please let us know if you have any more comments or questions. We would gladly try to address them.
>
> 1. "The results are shown for only one algorithm, DDQN, only on Atari. It would be good to provide results for newer algorithms like Ape-X or other environments like DMControl.":
>
> We tried to address your comment on DDQN and we added a new section in the Appendix on evaluation of policy gradient algorithms.
>
> 2. "Deep RL algorithms usually have large variance. It would be good to provide standard errors of the results, to accurately compare Daylight to Carlini & Wagner.":
>
> Yes of course. We also added this information in Table 1 through Table 6.
>
> 3."I think it would be better if the experimental setup preceded the results in Section 3. In addition, more details should be given, such as: How were the specific games and algorithm chosen?":
>
> We evaluated DDQN because it is known to be well performing and well established. Our aim was to provide a framework for any algorithm. We also added a new section on evaluation of policy gradient methods in Appendix Section A.3.
>
> Games are chosen based on the semantic differences between the tasks learned,  the diversity in their action set sizes, and the perceptual dissimilarity of the states. For action set size and semantic differences please see Bellemare et. al. (2013). For a small example of semantic differences and similarities BankHeist ([BankHeist-v0](https://gym.openai.com/envs/BankHeist-v0/)) and Amidar ([Amidar-v0](https://gym.openai.com/envs/Amidar-v0/)) can be categorized as relatively more similar based on the semantics of the learned tasks and the perceptual similarity of the states.
>
> Marc G Bellemare, Yavar Naddaf, Joel Veness, and Michael. Bowling. The arcade learning environment: An evaluation platform for general agents. Journal of Artificial Intelligence Research., pp. 253–279, 2013.
>
>
> 4. "I think the paper [1] is relevant for related work, as it shows similar distortions are good attacks for image classification.":
>
> Thank you for the reference. We added the reference.
>
> 5. "The notation D(s) at the bottom of page 2 is undefined.":
>
> Thank you for pointing this out. We fixed this typo.
>
>  6. "What happens if we combine several of these attacks? Would that lead to even greater impact?":
>
> Yes, one can try to utilize different types of modifications in the same state with the aim of minimizing the perceptual similarity and maximizing the impact on the agent. Since the results reported in the paper for most of the agents do already have very high impacts, adding additional modifications may not change the results by much. However, for the ones which do not have significantly high impact one can try to utilize multiple modification methods as you suggested and as has also been further discussed in Dong et al. (2019).
>
> Yin, Dong, Raphael Gontijo Lopes, Jon Shlens, Ekin Dogus Cubuk, and Justin Gilmer. "A fourier perspective on model robustness in computer vision." Advances in Neural Information Processing Systems 32 (2019): 13276-13286.
>
>
> 7. "why are the results for compression artifacts more mixed?":
>
>
> The reason that compression artifacts have comparable impact with Carlini & Wagner (2017) could be possibly due to the frequency band they occupy in the Fourier Domain. Figure 1 shows that both Carlini & Wagner (2017) and compression artifacts modify a similar portion of the high frequency band.
>
> Thank you for the further questions and thoughtful comments. We have tried to address them. If you have any further questions please let us know. We would be glad to address them.

---

> > ### Author Response · Authors · 2020-11-22
> > **Thank you for the Review**
> >
> > We thank you for your in depth review, and finding our black-box attack , intuitively meaningful, computationally cheap, and realistic. We have added a section in Appendix A.3 to address your comments and revised the paper with respect to your suggestions. We hope that our response addresses your questions and comments. Please let us know if you have any further questions. We would be happy to address them.

---

### Official Review · AnonReviewer3 · 2020-10-28
**Interesting study on generalization of RL using realistic perturbations**

**Rating:** 6
**Confidence:** 4

**Review:**

Summary:

This work focuses on assessing the generalization of deep reinforcement learning by applying semantically meaningful perturbations to the RL agent’s observation system, e.g., brightness, blurring, contrats, shifting etc. The authors demonstrate strong degradation in the performance of RL agents in various Atari games, despite the semantic perturbations having significantly lower perceptual distance as opposed to planned adversarial attacks.


Review:

The paper is well-written and easy to understand. Related work has been adequately referenced and the paper is of significant interest to the ICLR community.


Strengths:

1. The evaluation is quite comprehensive and covers multiple Atari environments.

2. The chosen perturbations are semantically meaningful and cover a broad range of frequencies in the fourier domain.

2. The degradation impact from semantically meaningful perturbations has also been compared to that from adversarial attacks, while at the same time characterising their perceptual similarity to original observations.


Weaknesses:

1. My major concern is that the metric defined to capture Impact (eq 2 in section 2.5) is somewhat unclear. Normally, one defines a score and then normalizes it with constants "max" and "min", for instance, like (max - score)/(max - min). In such a case, "max" and "min" are generally constant upper and lower bounds on score. However, while at the surface eq 2 seems to be doing the same, it might actually be doing something different. This is because of how Score_{max} and Score_{min} are defined. Firstly, Score_{max} and Score_{min} are not constants but differ for an agent depending on its Q-function. Secondly, the Score_{min} term seems somewhat misleading. The authors state that "Score_{min} is the score at the end of the episode achieved by the agent who takes the action that minimizes its Q(s, a) function in every state visited". Since, Q(s,a) is the value of action a in state s given that the agent maximizes the value from the next state onwards, just choosing the minimizing action in the current state may not be akin to choosing the actual minimum return from an episode. Hence, this functional form used to capture Impact may be doing more than just a score normalization. Can the authors please comment on why they chose the definition of Impact as such and if this could be having potential impacts on the results shown throughout the paper?

2. Additionally, since the authors have conditioned the definition of Impact on Q-values, it seems like the framework is only restricted to evaluation of agents which learn a Q-function. Direct policy-based agents may not be amenable to the Daylight framework. Is it possible to extend the framework to handle RL agents, which do not learn a Q-function?

I am happy to reconsider my score if the above two major weaknesses can be addressed.

Minor comments:

1. In most experiments from section 3, BankHeist stands out as an exception in the sense that Carlini & Wagner (2017) seems to be competitive on it as opposed to the proposed perturbations. It would be interesting to analyze and understand what sets this game apart.

2. The text and labels on the axes in figure 1 are unreadable. Please rectify the font size.

3. I might have misunderstood this, but why does figure 1 have only one spatial frequency axis? Shouldn't it also have two spatial frequency axes like in figure 2?

4. Minor punctuation error in the abstract: "state-of-the-art adversarial attacks, ."

------ Post-rebuttal update ------

The reviewers have addressed my concerns for the most part and I am happy to update my score to recommend acceptance. I also hope to see the A3C experiments in the final draft of the paper.

---

> ### Author Response · Authors · 2020-11-16
> **Author Response**
>
> Thank you for your insightful comments. We have tried to address them below. Please let us know if you have any more questions or comments that you would like us to address. We would be glad to address them.
>
> 1. “My major concern is that the metric defined to capture Impact (eq 2 in section 2.5) is somewhat unclear.”:
>
>
> You make a good point that there may be subtle issues when using Impact to compare agents trained with different algorithms, due to the dependence on the agent’s Q-function. The reason we reported the values as defined in Equation 2 Section 2.5 is that we were making a comparison to the state-of-the-art Carlini-Wagner adversarial attack. When using the Carlini-Wagner formulation to cause the agent to choose a non-optimal action in each state, we thought a somewhat natural baseline would be to deterministically force the agent to take the worst possible action in each state (as evaluated by the Q function).
>
> To address your concerns we added tables reporting the raw scores for all of our perturbations in Appendix A.1. Based on your suggestions we also defined a generalized Impact across different algorithms with a fixed value for $Score_{min}$ given an environment and reported its values in Appendix A.2. These results are also consistent with the scores as measured by the Impact metric in Equation 2 Section 2.5.
>
> 2. “Since, $Q(s,a)$ is the value of action a in state s given that the agent maximizes the value from the next state onwards, just choosing the minimizing action in the current state may not be akin to choosing the actual minimum return from an episode.”
>
> It is definitely true that choosing the minimizing action in the current state may not get the actual minimum returns possible in an episode. We tried to address your concerns with the Impact definition in Appendix A.1 and A.2 as described above.
>
> 3. “Is it possible to extend the framework to handle RL agents, which do not learn a Q-function?”:
>
> Yes of course. We added a section dedicated to this in Appendix A.2 and A.3.
>
> For the scope of the paper we used the Impact definition in Equation 2 Section 2.5 when we compare our proposed Daylight framework to the state-of-the-art targeted attacks. For more generalized comparison between different algorithms and different games one can use $Score_{clean}$ as the score of the agent without any modification to agent's observations system at the end of the episode and $Score^{fixed}_{min}$ as the fixed minimum score for a given game. Thus, we define the generalized impact as,
>
> $I_{general} = (Score_{clean}-Score_{adv})/(Score_{clean}-Score^{fixed}_{min}).$
>
> 4. "I might have misunderstood this, but why does figure 1 have only one spatial frequency axis? Shouldn't it also have two spatial frequency axes like in figure 2?":
>
> The x-axis in Figure 1 is the maximum of the magnitude of the two spatial frequencies. We chose to plot it this way so as to be able to clearly visualize the changes in the spectrum without relying on color-based heatmaps in which the comparison might be more difficult to see.
>
> Thank you for the detailed comments and insightful thoughts. We have tried to address them. Please let us know if you have any further questions. We would be glad to answer them.

---

> > ### Author Response · Authors · 2020-11-22
> > **Thank you for the Review**
> >
> > We added sections in Appendix A.1, A.2 and A.3 to address your concerns. We hope that our response and added sections addressed your questions. Please let us know if you have any other comments or questions. We thank you again for your helpful suggestions and insightful review.

---

> > ### Comment · AnonReviewer3 · 2020-11-24
> > **Thank you for the clarifications**
> >
> > Thank you for providing further clarifications and adding new results with the generalized impact definition. While the extension to policy gradient based methods (A3C) in appendix A.3 is very short and not informative, the other results in appendices A.1 and A.2 look acceptable to me. I will update my score after further discussion with other reviewers during the discussion period.

---

> > > ### Author Response · Authors · 2020-11-24
> > > **Thank you for your response**
> > >
> > > Thank you very much for your response.  Is there anything we can add to make Appendix Section A.3 more informative? Do you think we should add impacts, and generalized impacts? We just wanted to add this initial result on A3C to show that the vulnerabilities we observed in the paper are not specific to DDQN.

---

> > > > ### Comment · AnonReviewer3 · 2020-11-24
> > > > **Response**
> > > >
> > > > Yes, having the impact values and the generalized impact values would help a lot. Also, section A.3 only reports values for Pong currently. It would be great to see the numbers for other games too.
> > > >
> > > > PS: I understand that the experiments on other games could take some time to run and that the discussion period ends today, so I don't expect to see them today but hopefully in a next update.

---

> > > > > ### Author Response · Authors · 2020-11-24
> > > > > **Response**
> > > > >
> > > > > Thank you very much for your prompt response. We will add the impact values just now. Thank you so much for your understanding on the time required for the experiments. We will try to make them ready for the camera ready version in case of acceptance.

---

### Official Review · AnonReviewer2 · 2020-10-29
**More experimental results and better evaluation of generated samples are needed**

**Rating:** 5
**Confidence:** 3

**Review:**

Strong points:
1. The authors propose a new framework for constructing adversarial data samples for deep reinforcement learning training. It contains six different concrete methods. Some of them can generate data samples fairly different from the original data samples.
2. The authors do experiments with the proposed technique to check the generalizability on 10 atari games, and show that the generated data samples can fail the agent training to a different extent.

Weak points:
1. The biggest problem is that: although the authors have demonstrated that the generated data samples are fairly different from the original data samples, how to guarantee that there is still enough meaningful information contained in these training data samples? Definitely, we can look at the generated figures and it seems that they are still recognizable. However, we probably need some quantitative measures to show that these data samples can still be used to train good agents. To the extreme, we can enumerate some random samples, which will be extremely different from the original samples, and they will also fail the training of the rl agents.
2. More quantitative experimental results are needed. It is more convincing to see how much score loss can the proposed framework cause, compared with clean data samples.
3. One central part of the evaluation should aim to answer: whether this framework can differentiate the generalizability of algorithms better than other frameworks. This should be shown in experiments. For instance, say we have three algorithms, A, B and C, and we believe the groundtruth is their generalizability is A >> B ~ C.  Then, under this framework, we should see the score align with this conclusion better than under other frameworks.

---

> ### Author Response · Authors · 2020-11-16
> **Author Response**
>
> Thank you for your comments. We have tried to address them below. Please let us know if you have more comments and questions. We would gladly try to address them.
>
> 1. “The authors propose a new framework for constructing adversarial data samples for deep reinforcement learning training. It contains six different concrete methods. Some of them can generate data samples fairly different from the original data samples.”:
>
> Our daylight framework is for testing the trained agents in various modifications to the observations of the agent to measure the robustness of the models. In particular, the agents are trained with clean data and their performance is tested in a run of the game where the modifications are applied to the agent’s observations.
>
> 2. “The biggest problem is that: although the authors have demonstrated that the generated data samples are fairly different from the original data samples, how to guarantee that there is still enough meaningful information contained in these training data samples? Definitely, we can look at the generated figures and it seems that they are still recognizable. However, we probably need some quantitative measures to show that these data samples can still be used to train good agents. To the extreme, we can enumerate some random samples, which will be extremely different from the original samples, and they will also fail the training of the rl agents.”:
>
> We already indeed provide this quantitative measure in Table 1 through Table 6. The quantitative measure used is the Learned Perceptual Image Patch Similarity (LPIPS) metric as described in Section 2.4. You are right to wonder about this quantitative measure, as without such a measure it would have been vague to claim  similarities between the original observation (unmodified) and the modified observations. However, we did already provide this data in Section 2.4 and Table 1 through Table 6.
>
> 3. "It is more convincing to see how much score loss can the proposed framework cause, compared with clean data samples.":
>
> This is also already provided in the paper by Impact values in Table 1 through Table 6. The main quantitative data that we report for our framework is the Impact metric defined in Section 2.5. This metric is a normalized measure of how much the raw score of the agent drops when compared to an agent run on clean data samples. We believe that this provides a good comparison of the score loss between our framework and clean data. But we also added the raw scores in Appendix A.1. If you prefer a different metric for measuring score loss we would be happy to include it in the paper. Please let us know if you have any more questions on this.
>
> 4. "One central part of the evaluation should aim to answer: whether this framework can differentiate the generalizability of algorithms better than other frameworks. This should be shown in experiments. For instance, say we have three algorithms, A, B and C, and we believe the groundtruth is their generalizability is A >> B ~ C. Then, under this framework, we should see the score align with this conclusion better than under other frameworks.":
>
> One of the frameworks for robustness evaluation is based on investigation through adversarial examples. Currently the Carlini & Wagner (2017) adversarial formulation is the state-of-the-art adversarial formulation in the community [Athalye et. al (2018); Sun et al. (2020); Lin et al. (2017)]. We also provide impact and perceptual similarity results for the adversarial evaluation framework (Carlini &Wagner (2017)) and the  daylight framework. These results are in Table 1 through Table 6. Furthermore, we compare the adversarial evaluation framework and the daylight framework in the frequency domain and provide an analysis on this in Section 4. Moreover, we investigate the question of  timing in observation manipulation between the adversarial evaluation framework and the daylight framework in Section 5.
>
> Thank you very much for your comments. If you have any more comments on the paper or how the paper is organized please let us know. We would be glad to answer them.
>
>
> Nicholas Carlini and David Wagner. Towards evaluating the robustness of neural networks. In 2017 IEEE Symposium on Security and Privacy (SP), pp. 39–57, 2017.
>
> Jianwen Sun, Tianwei Zhang, Lei Xiaofei, Xie Ma, Yan Zheng, Kangjie Chen, and Yang. Liu. Stealthy and efficient adversarial attacks against deep reinforcement learning. Association for the Advancement of Artificial Intelligence (AAAI), 2020.
>
> Yen-Chen Lin, Hong Zhang-Wei, Yuan-Hong Liao, Meng-Li Shih, ing-Yu Liu, and Min Sun. Tactics of adversarial attack on deep reinforcement learning agents. Proceedings of the Twenty-Sixth International Joint Conference on Artificial Intelligence, pp. 3756–3762, 2017.
>
> Anish Athalye, Nicholas Carlini, David A. Wagner:Obfuscated Gradients Give a False Sense of Security: Circumventing Defenses to Adversarial Examples. ICML 2018: 274-283

---

> > ### Author Response · Authors · 2020-11-22
> > **Thank you for the Review**
> >
> > We hope that our response addressed your concerns and questions. Please let us know if you have any other questions about the paper. We would be happy to answer them.

---

> ### Author Response · Authors · 2020-11-25
> **Discussion Period**
>
> The discussion period is now ending. Thank you for your time for your initial review. We hope that our response addressed your questions and comments, and that it will be taken into consideration in your review update.

---

### Author Response · Authors · 2020-11-24
**Author Meta Response**

Dear Reviewers,

We thank the reviewers finding our work comprehensive, insightful, and of significant interest for the ICLR community and finding our approach semantically meaningful, computationally cheap, and realistic.

We thank all of you for your detailed comments and insightful questions. We tried to address your reviews individually. For the most convenience we also add a detailed summary of the revisions made during the author rebuttal below:


* **[16.11.2020]** Raw Scores of the agents under observation modifications for Carlini \& Wagner (2017) and Daylight framework have been added in Appendix Section A.1.

* **[16.11.2020]** **Fixed Impact** has been defined to compare the generalization capabilities of different DRL algorithms. Fixed impacts of the agents under observation modifications for Carlini \& Wagner (2017) and Daylight framework have been added in Appendix Section A.2.

* **[16.11.2020]** Policy gradient evaluation has been added to Appendix Section A.3.

* **[16.11.2020]** Sample standard deviations are added in Table 1 through Table 6. We reported the sample standard deviations to give a measure of how spread out the distribution of impacts is, but we can also add the standard error of the mean impact if the reviewers think this metric would provide more information.

* **[24.11.2020]** Generalized impacts and more details are added to Appendix Section A.3.

---

### Decision · Program_Chairs · 2021-01-07
**Final Decision**

**Decision:**

Reject

**Comment:**

The paper proposes several simple alternatives to generate adversarial examples for deep reinforcement learning algorithms based on image distortions such as lighting change, blurring and rotation, and show the performance of DRL agents degenerate as a result. Most reviewers appreciate the simplicity and computational efficiency of the proposed attacks. The results revealed by the work is however rather unsurprising, given similar attacks have been evaluated for DNNs. The authors did not offer much more insight on the presented results beyond that, for example, robustness of different DRL algorithms with regards to these attacks as mentioned by reviewer 2, sensitivities of the parameters for each attack proposed, effectiveness of different attacks on different environment and possible combination of attacks.